# Targeting VIP and PACAP Receptor Signaling: New Insights into Designing Drugs for the PACAP Subfamily of Receptors

**DOI:** 10.3390/ijms23158069

**Published:** 2022-07-22

**Authors:** Jessica Lu, Sarah J. Piper, Peishen Zhao, Laurence J. Miller, Denise Wootten, Patrick M. Sexton

**Affiliations:** 1Drug Discovery Biology, Australian Research Council Centre for Cryo-Electron Microscopy of Membrane Proteins, Monash Institute of Pharmaceutical Sciences, Monash University, Parkville, VIC 3052, Australia; jessica.lu@monash.edu (J.L.); sarah.piper@monash.edu (S.J.P.); elva.zhao@monash.edu (P.Z.); 2Department of Molecular Pharmacology and Experimental Therapeutics, Mayo Clinic, Scottsdale, AZ 85259, USA; ljm@mayo.edu

**Keywords:** pituitary adenylate cyclase-activating polypeptide (PACAP), vasoactive intestinal peptide (VIP), receptor selectivity, VIP receptor 1 (VPAC1R), VIP receptor 2 (VPAC2R), PACAP type I receptor (PAC1R), peptide therapeutics, structure-based drug design, GPCR

## Abstract

Pituitary Adenylate Cyclase-Activating Peptide (PACAP) and Vasoactive Intestinal Peptide (VIP) are neuropeptides involved in a diverse array of physiological and pathological processes through activating the PACAP subfamily of class B1 G protein-coupled receptors (GPCRs): VIP receptor 1 (VPAC1R), VIP receptor 2 (VPAC2R), and PACAP type I receptor (PAC1R). VIP and PACAP share nearly 70% amino acid sequence identity, while their receptors PAC1R, VPAC1R, and VPAC2R share 60% homology in the transmembrane regions of the receptor. PACAP binds with high affinity to all three receptors, while VIP binds with high affinity to VPAC1R and VPAC2R, and has a thousand-fold lower affinity for PAC1R compared to PACAP. Due to the wide distribution of VIP and PACAP receptors in the body, potential therapeutic applications of drugs targeting these receptors, as well as expected undesired side effects, are numerous. Designing selective therapeutics targeting these receptors remains challenging due to their structural similarities. This review discusses recent discoveries on the molecular mechanisms involved in the selectivity and signaling of the PACAP subfamily of receptors, and future considerations for therapeutic targeting.

## 1. Introduction—Physiological Roles of the PACAP and VIP Receptors

Pituitary Adenylate Cyclase-Activating Peptide (PACAP) and Vasoactive Intestinal Peptide (VIP) are neuropeptides involved in a multitude of physiological and pathological processes. VIP is a 28 amino acid peptide that is widely expressed in various cell types in the central (CNS) and peripheral nervous systems [1]. Centrally, VIP promotes neuronal survival [2] and regulates glycogen metabolism in astrocytes [3], while peripherally, VIP functions as a vasodilator leading to non-adrenergic and non-cholinergic relaxation of vascular and non-vascular smooth muscle through co-transmission with nitric oxide and carbon monoxide [4,5]. VIP is also co-transmitted with acetylcholine in exocrine glands [5,6,7], and works as a secretagogue by stimulating prolactin secretion in the pituitary and catecholamine release in the adrenal medulla [8,9]. In the immune system, VIP is an immunoregulator that controls T-lymphocyte trafficking and inhibits interleukin-2 (IL2) production [10]. VIP also stimulates electrolyte secretion in the jejunum [11,12], and plays a protective role against oxidant injury [13].

Similar to VIP, PACAP is widely expressed and is involved in numerous physiological processes, mediated both centrally and peripherally [14,15,16]. PACAP exists in two isoforms from proteolysis of the same precursor protein in the body, PACAP38 and PACAP27, which differ by eleven amino acid residues in the C-terminus [17,18]. Both PACAP38 and PACAP27 are bioactive, however, PACAP38 is more than 100-fold more abundant than PACAP27 in neuronal tissues [14,15]. PACAP plays important roles in the CNS as a neurotransmitter, being involved in learning and memory [19], and control of circadian rhythms [20]. Additionally, PACAP exerts a neuroprotective effect in responses to stress [21], including during brain injury and ischaemia [22,23]. Peripherally, PACAP modulates control of vasodilation [24,25], and secretion of insulin [26], adrenaline and anterior pituitary hormone [27,28].

PACAP and VIP exert their physiological effects by activating the PACAP subfamily of G protein-coupled receptors (GPCRs) that consist of VIP receptor 1 (VPAC1R), VIP receptor 2 (VPAC2R) and the PACAP type I receptor (PAC1R). These receptors are distributed in the central and peripheral nervous systems, endocrine system, and immune system, and mediate the physiological effects of the peptides described above, including control of circadian rhythm [29,30,31], immune regulation [32,33], and insulin secretion [34]. A summary of the distribution and physiological or potential therapeutic roles of these receptors is described below (Table 1).

Due to their extensive, yet discrete, distribution and diverse physiological roles, the PACAP subfamily receptors are attractive targets for the development of therapeutics, particularly for the treatment of chronic inflammation, nociceptive pain, and neurodegeneration (Table 1) [33,85,86].

The PAC1R is involved in the regulation of pleiotropic neurological functions including promotion of neural survival and synaptic plasticity, nociceptive pain, and regulating the hypothalamic–pituitary–adrenal axial stress response, which makes it an attractive target for nervous system disorders including migraine [85,102,103], secondary injury in traumatic brain injury [84], and post-traumatic stress disorder [81,83]. The PAC1R is expressed in regions of the trigeminal-autonomic system that are associated with migraine, namely the sphenopalatine autonomic ganglia neurons and spinal trigeminal nucleus [104]. PAC1R activation mediates intracranial nociceptive activation of the central trigeminal-vascular neurons and induces neurogenic dural vasodilation contributing to migraine pathophysiology [102,103]. PACAP/PAC1R signaling is also involved in the stress mechanisms of the body [105,106]. PACAP and PAC1R are highly expressed in stress-associated brain regions in the hypothalamic and limbic structures [14,73,82]. PAC1R signaling modulates corticosterone and corticotropin-releasing hormone levels [107,108], with PACAP and PAC1R knockout mice displaying reduced anxiety-like phenotypes and blunted corticosterone response [83,109]. In women, PACAP serum levels are directly correlated to PTSD symptoms [83]. These sex-specific effects are speculated to be associated with oestrogen-dependent regulation of PACAP systems within the bed nucleus of stria terminalis in the extended amygdala [83].

On the other hand, VPAC receptors are attractive targets for immune and inflammatory conditions. VPAC1Rs are expressed constitutively, or in the case of VPAC2Rs, following induction, in T-lymphocytes and macrophages in the immune system [53,54,55,56]. In homeostatic states, the VPAC1 and VPAC2 receptors play an important role in preserving the equilibrium of pro- and anti-inflammatory responses through the regulation of various inflammatory mediators in immune cells [53,54,56]. While in pathological inflammatory states, these receptors can promote anti-inflammatory outcomes by inhibiting the pro-inflammatory Th1 and Th17 responses and stimulating Th2 and Treg responses [110,111,112]. As such, there is significant interest in targeting these receptors for the treatment of chronic inflammatory diseases including Crohn’s [113], Sjogren’s syndrome [114], COPD [45], autoimmune encephalitis [115], and rheumatoid arthritis [116].

## 2. Pharmacology of PACAP and VIP Receptors

To coordinate the diverse physiological outcomes mediated by the activation of the PACAP subfamily receptors by their cognate peptide ligands, these receptors activate multiple signaling pathways through the coupling of different G proteins, as well as through the interactions with a myriad of scaffolding and accessory proteins to stimulate additional signaling pathways.

### 2.1. Binding Characteristics of Natural Peptides Differentiating the PACAP Subfamily of Receptors

The PACAP subfamily receptors were initially differentiated by their distinct peptide binding affinity profiles. Early studies in tissues using ^125^I-labelled PACAP identified two types of PACAP receptors. The first type, which was classified as the PACAP type I receptor or PAC1R, had a high binding affinity for PACAP (K_d_ of approximately 0.5 nM) with a lower affinity for VIP (K_d_ > 500 nM) [117,118,119,120]. In contrast, the second type of PACAP receptors had a similar affinity (K_d_ of approximately 0.5 nM) for both VIP and PACAP and were subsequently classified as PACAP type II receptors [14,118,119]. Later, it was discovered that the type II PACAP receptor subclass contained two distinct types of receptors [121,122]. One (later named VPAC1R) had a lower affinity for helodermin, an exogenous peptide derived from the venom of the lizard *Heloderma suspectum* [123,124], compared to VIP and PACAP [125], and the other (VPAC2R) had a similar affinity toward all three peptides [59,126].

Other peptide agonists of the VPAC receptors were later identified, including peptide histidine isoleucine (PHI) in rodents and peptide histidine methionine (PHM) in humans, which are derived from post-translational processing of the VIP precursor following mRNA translation and were revealed to be biologically active [127,128,129]. These peptides share almost 50% sequence homology to VIP and bind with lower affinity to the VPAC1R and VPAC2R receptors compared to VIP [130,131]. Maxadilan, a PAC1R selective agonist that was originally isolated from the sand-fly (*Lutzomyia longipalpis*) salivary gland [132], shares no significant sequence similarity to VIP and PACAP, but can also bind and activate the PAC1R with a similar affinity and potency to PACAP [133,134].

#### Selective Peptide Analogues of PAC1R, VPAC1R, and VPAC2R

Several of the natural peptides described above are able to interact with multiple members of the PACAP receptor subfamily. However, non-selective binding may lead to undesirable side effects in clinical practice, hence the development of receptor-selective ligands may improve therapeutic utility. Before the determination of the PACAP-bound VPAC1R, VPAC2R, and PAC1R structures through cryo-electron microscopy (cryo-EM) [135,136,137,138,139], peptide analogues were designed and generated through systematic residue scanning of the peptide sequences seeking improved receptor selectivity. These analogues were subsequently screened in in vitro assays for binding and receptor signaling. From this process, several peptide analogues were identified that exhibited improved selectivity within the PACAP subfamily of receptors (Table 2).

By categorizing receptor-selective analogues by their modifications to the natural peptide, several patterns emerge that provide a foundation for structure-activity relationship analysis and drug design. Firstly, deletion or modification of the N-terminal residues in VIP and PACAP analogues can be used to generate antagonists as seen with PACAP (6-38), PG 97-269, and PG 99–465 [145,147,149,152] (Table 3). As observed in general for endogenous Class B1 GPCR peptides, the N-terminal region is essential for receptor activation. The N-terminal truncation of multiple endogenous peptides of class B1 GPCRs, including corticotropin-releasing factor (CRF), the CRF analogue urotensin, the glucagon-like peptide-1 receptor (GLP-1R) agonist exendin, calcitonin, parathyroid hormone (PTH)-related peptide (PTHrP), glucagon and secretin results in the production of antagonists to their respective receptors [153,154,155,156,157,158,159].

Secondly, at the C-terminal end, modifications to peptide length play a role in VPAC1R/VPAC2R selectivity (Table 4). RO 25-1553 was initially developed as a long-acting, VPAC2R selective, VIP analogue to overcome some of the deficiencies in the natural peptide for clinical bronchodilatory action; namely the short half-life in vivo due to Thr^7P^-Asp^8P^ and Ser^25P^-Ile^26P^ cleavage. The VIP analogue RO 25-1553 introduced a lactam bridge between the amino acid residues 21 and 25, along with an elongated C-terminal tail (Asp^25P^-Leu^26P^-Lys^27P^-Lys^28P^-Gly^29P^-Gly^30P^-Thr^31P^), generating a peptide with a long duration of bronchodilatory activity, compared with the natural peptide [160]. PG 96-249a, a RO 25-1553 analogue where the lactam bridge between positions 21 and 25 is absent, retained VPAC2R selectivity [147], suggesting that the elongation of the C-terminus rather than the lactam bridge was involved in VPAC2R selectivity. This improvement in VPAC2R selectivity with C-terminal peptide length is also observed when extending the C-terminal end of a VIP analogue from 28 to 31 residues with the equivalent PACAP38 residues [161]. On the other hand, [Arg^16^]-PACAP (1–23), a C-terminally truncated analogue was selective for VPAC1R, while the full-length [Arg^16^]-PACAP (1–27) was non-selective between VPAC1 and VPAC2R [143].

Thirdly, when analyzing common residues where modifications alter receptor selectivity, position 22 on the C-terminal end of VIP appears to play a role in VPAC1R/VPAC2R selectivity [141]. Alanine scanning revealed that the aromatic Tyr residue at position 22 was important for high-affinity VPAC2R binding, while it was not as crucial for VPAC1R binding [141]. These empirical observations in conjunction with emerging structural data provide guidance for future peptide and potentially small molecule drug design for VIP and PACAP receptor selective therapeutics.

### 2.2. Signaling Characteristics of PAC1R, VPAC1R, and VPAC2R

Similar to other class B1 GPCRs, the PACAP family receptors predominantly couple to G_s_ protein to activate the adenylate cyclase (AC) to produce cAMP (3′,5′-cyclic adenosine monophosphate) (Figure 1). However, some members of this receptor family also couple to G_q/11_ and G_i/o_ proteins to induce activation of phospholipase C (PLC) and intracellular calcium (Ca^2+^) mobilization [162,163,164,165]. Many class B1 GPCRs also signal through scaffolding proteins such as β-arrestin and A-kinase anchoring proteins (AKAPs) [166,167]. Furthermore, certain receptors in this family can associate with receptor activity-modifying proteins (RAMPs) to modulate receptor signaling and trafficking [168,169,170,171,172].

#### 2.2.1. Alternative Splicing

Some class B1 receptors, including the CRF, calcitonin, and PTH1R, undergo alternative splicing events resulting in receptor variants that can exhibit diverse signaling outcomes [173]. Alternative splicing is a regulatory mechanism during the pre-mRNA splicing process that results in the production of multiple mRNA variants from a single gene [174]. PAC1R undergoes extensive alternative splicing resulting in a host of receptor variants known as splice isoforms [175,176,177,178,179], while reports of this phenomenon occurring in the VPAC receptors are limited [173,180]. In humans, these splice isoforms occur in the N-terminal extracellular domain (ECD) and/or the intracellular loop 3 (ICL3) of PAC1R with the most prevalent and common variant containing a full-length N-terminal ECD and no ICL3 insertions (PAC1R null, a.k.a. PAC1nR) (Figure 2) [176,181].

Splice isoforms involving the N-terminal ECD including the variant with a truncation within the ECD, PAC1R short (PAC1sR), can affect ligand-binding specificity and affinity, while isoforms of ICL3 may alter G protein coupling and receptor trafficking, enriching complexity in PAC1R signaling [175,178,182,183,184,185]. Splice isoforms from alternative splicing events at ICL3 are characterized by the inclusion of one or two of the cassette exons, exons 14 (the “hip” cassette) and 15 (the “hop” cassette) of 28 amino acids each [180]. The inclusion of the hip cassette in PAC1R leads to the splice isoform PAC1R-hip [176]. Inclusion of the hop cassette leads to the PAC1R-hop, while the inclusion of both hip and hop cassettes leads to PAC1R-hiphop [181,186]. While these splice variants may have distinct pharmacological profiles, the physiological implications of these have not been studied in detail.

A summary of the expression of the PAC1R splice isoforms in the nervous system is published in Blechman et al. [180]; the spatiotemporal differences in isoform expression during embryonic and postnatal development suggest a role of the splice isoforms in the regulation of neurodevelopmental processes. While limited studies have characterized PAC1R splice isoform expression in humans, studies characterizing isoform expression in other species during development have been performed. In studies in rats, the hop, hiphop, and short PAC1R splice isoforms were expressed in the developing brain with an expression of the hop and short isoform increasing from embryonic days 10 to 21 and the hiphop variant being detected from embryonic day 17 to birth [89]. Rat cortical expression of the hip, hop and hiphop PAC1R splice variants was highest immediately after birth with expression levels decreasing significantly one month later [187]. In the rat retina, expression of the null and hip isoforms was the highest immediately after birth, with expression decreasing overtime suggesting that these isoforms might be involved in early stage retinal development, while the hop isoform expression increased at day 10–20 postnatal suggesting a role in later-phase retinal development [188]. Interestingly in both these studies, the expression of VPAC1R and VPAC2R increased over time with cortical expression increasing significantly one month postnatal and retinal expression peaking at days 10–15 after birth for VPAC1R and days 5–15 after birth for VPAC2R [187,188]. Additionally, the PAC1-hop isoform was induced in response to stress in zebrafish [189]. An age-dependent effect in hop isoform-deficient zebrafish was observed, where the hop isoform-deficient larvae exhibited increased anxiety-like behavior, while the adult hop isoform-deficient zebrafish exhibited the opposite response to novelty stress, suggesting a role of this PAC1R isoform in stress adaptation response development [190]. However, species differences must be taken into account when extrapolating these findings to humans. In one study, where isoform expression in the rat and human placenta was studied, expression of the short, the hip or hop and the, hiphop PAC1R isoforms were detected in the rat placenta, while only the hop isoform was detected in the human placenta [191].

#### 2.2.2. Signaling via G_s_ Coupling

All three receptors in the PACAP subfamily couple to the G_s_ protein resulting in the activation of AC and production of cAMP (Figure 1C) [146,178,192]. cAMP is a nucleotide that acts as a second messenger in various signaling pathways. cAMP can activate numerous effector proteins including protein kinase A (PKA) and exchange proteins activated by cAMP (EPAC).

PAC1R-induced cAMP production is associated with broad roles in the nervous system including the promotion of neuronal survival, regulation of cerebellar development, neuritogenesis, and the enhancement of N-methyl-D-aspartate (NMDA) receptor activity through PKA signaling [193,194,195,196,197]. PAC1R-induced cAMP activation of EPAC has been associated with neuronal excitability and differentiation through activation of Rap1 [198,199]. In addition, PAC1R-induced cAMP signaling is associated with neuronal differentiation independently of PKA and EPAC activation through an effector termed the neuritogenic cAMP sensor [200,201]. This signaling pathway involved the activation of extracellular signal-regulated kinase (ERK) by the neuritogenic cAMP sensor [200].

For the VPAC receptors, VPAC2R signaling by cAMP/PKA promotes circadian clock gene expression through cAMP-response element binding (CREB) phosphorylation [202], enhances synaptic transmission to hippocampal CA1 pyramidal cells [203], and through activating ERK phosphorylation, regulates anterior pituitary cell secretion and proliferation [204]. In the immune system, VPAC1R inhibits TNFα production through cAMP/PKA activation in macrophages [205], and is associated with pro-inflammatory exocytosis of secretory vesicles and granules from monocytes through cAMP/EPAC activation [206].

#### 2.2.3. Signaling via G_q/11_ and G_i/o_ Coupling

All three receptors of the PACAP subfamily also couple to G_q/11_ resulting in the activation of PLC (Figure 1C) [178,192,207]. PLC cleaves phosphatidylinositol 4,5-bisphosphate to form diacylglycerol (DAG) and inositol 1,4,5-trisphosphate (IP_3_). In turn, DAG activates phosphokinase C (PKC), while IP_3_ stimulates intracellular Ca^2+^ release from the endoplasmic reticulum [177,180,208].

G_q/11_ coupling plays an important role in PAC1R-modulated NMDA receptor activity and neuroplasticity in the CNS. While both PAC1R-mediated G_s_/cAMP/PKA signaling and PKC signaling through G_q/11_ are involved in the enhancement of NMDA receptor-mediated activity, PLC knockout resulted in a major reduction in the PAC1R-activated NMDA receptor-mediated excitatory postsynaptic currents despite a similar level of cAMP/PKA activation, suggesting G_q/11_/PKC signaling is the dominant signaling pathway in modulation of NMDA receptors in hippocampal CA1 neurons [196,209]. PAC1R/PKC signaling can also activate ERK pathways for adaptative neuronal responses [210,211]. Additionally, in rat cerebral cortical slices, PAC1R stimulates PLC-IP_3_/DAG signaling in astrocytes [209,212].

While PAC1R activation of PLC has thus far been reported to exclusively involve G_q/11_-mediated signaling, VPAC1R and VPAC2R can interact with both G_i/o_ and G_q/11_ proteins to elicit PLC activation [207,213]. However, this response appears to be species and cell-type dependent. VPAC1R couples to G_i3_ in rat alveolar macrophages [214], while PLC activation is exclusively mediated through G_q/11_ by human VPAC1R and VPAC2R when overexpressed in COS-7 cells [207]. In gastric muscle cells, VPAC2R stimulates nitric oxide synthase activity through G_i1-2_ [215]. While in the pancreas, VPAC2R activates PLC and mobilizes intracellular Ca^2+^ through a G_i_-activated mechanism [216].

#### 2.2.4. Non-G Protein Signaling

In addition to direct G protein-mediated signaling, the VPAC receptors and certain variants of the PAC1R can signal via G protein-dissociated mechanisms [217,218,219].

##### ADP-Ribosylation Factor

The VPAC1R and VPAC2R stimulate phospholipase D (PLD) activation through ADP-ribosylation factor (ARF) (Figure 1A) [217]. Moreover, the PAC1R stimulated PLD exclusively through the PAC1nR-hop1 variant [217,218]. The physiological significance of this pathway in the PACAP subfamily of receptors is yet to be established however, PLD activation has been associated with physiological responses including a respiratory burst in neutrophils, regulation of transport and endocytosis, and changes in cell morphology and motility [220,221].

##### Endosomal Signaling

β-arrestins can also provide a means for G protein-dissociated signaling by acting as scaffolds to facilitate multiple interactions between GPCRs and cytoplasmic signaling proteins for initiation of endosomal signaling [222]. Endosomal signaling by the PAC1R can provide distinct spatial and temporal control of cellular signaling (Figure 1B). In guinea pig cardiac neurons, PAC1R internalization and endocytosis modulate neuronal excitability via PKA/ERK activation [100,223,224,225], while PAC1R-activated endosomal ERK signaling in the amygdala regulates nociceptive hypersensitivity responses [75]. In the hypothalamus, PAC1R trafficking is involved in regulating feeding behavior and energy homeostasis [226]. In primary sympathetic neurons, the PAC1R splice variant, PAC1nR-hop1 can activate endosomal PI3k/Akt signaling cascades to facilitate neuronal survival following growth factor withdrawal [227]. While the PAC1R has been extensively documented to undergo additional signaling following receptor internalization, to date endosomal signaling has not been reported for the VPACRs [228].

#### 2.2.5. Additional Downstream Signal Transduction Pathways

##### Ion Channels

As described above, cytosolic Ca^2+^ concentrations can be altered by Ca^2+^ release through transporter proteins in the endoplasmic reticulum via PLC-dependent mechanisms. Additionally, the PACAP receptor subfamily can modulate intracellular Ca^2+^ concentrations through the activation of ion channels. Downstream of G protein-mediated PKA and PKC activation, these kinases can modulate ionic currents to control intrinsic neuronal excitability and function following PACAP and VIP receptor activation. The PAC1R also modulates extracellular Ca^2+^ influx following PKC activation via voltage-gated Ca^2+^ channels [229,230,231,232], while PAC1R/PKA signaling can inhibit potassium (K^+^) channels, promoting neuronal survival [193,194]. In hippocampal nerve terminals, VPAC1R inhibits voltage-gated Ca^2+^ channel-dependent GABA release through G_i/o_-mediated PKC activation, while VPAC2 enhances voltage-gated Ca^2+^ channel-dependent GABA release through G_s_-mediated PKA and PKC activation [233].

In the periphery, PAC1R-stimulated PKA signaling can activate Ca^2+^ channels and inhibit voltage-gated potassium (K^+^) channels leading to insulin secretion in pancreatic β-cells [234]. In addition, PAC1R/PLC signaling in PDGFRα+ interstitial cells may be involved in mediating the inhibitory regulation of colonic contractions through small conductance Ca^2+^-activated K^+^ channels 3 [235].

##### Transactivation of Epidermal Growth Factor Receptor (EGFR)

Through PKA signaling, the PAC1R also transactivates EGFR, resulting in the activation of ERK signaling that promotes cell survival and proliferation [96]. In diabetic retinopathy, this pathway improves cell survival following hyperglycaemic injury [96], while in growth factor-deprived human corneal endothelial cells, it promotes wound healing and improves corneal endothelial barrier integrity [93,236]. Activation of insulin-like growth factor-1 (IGF-1) in cortical neurons transactivates PAC1R, and this mechanism was important for its anti-apoptotic activity [236]. In cortical neurons, PAC1R transactivated EGFR through PKA signaling, however EGFR transactivation only had a minor contribution to the anti-apoptotic actions of IGF-1 [236]. On the other hand, PAC1R-mediated EGFR activation may promote the proliferation of certain lung cancer cells by PKC-dependent EGFR activation [237], and VPAC1 signalling can transactivate EGFR and human epidermal growth factor receptor 2 (HER2) through PKA and Src signalling in breast cancer cells [238].

##### MAPK/ERK Signal Transduction

Aside from modulating ion currents, receptor-mediated activation of the protein kinases, PKA and PKC, can activate ERK signaling. ERK belongs to a family of serine/threonine kinases known as mitogen-activated protein kinases (MAPK) that are important effectors of GPCRs that regulate cellular growth, division, differentiation, and apoptosis [239,240]. All three members of the PACAP receptor subfamily can activate ERK signaling via PKA activation [200,204,241]. However, PKA and PKC-independent ERK activation also occurs following PAC1R activation in astrocytes, promoting cell proliferation [242]. Moreover, MAPK/ERK signaling can be activated through both plasma membrane and endosomal signaling pathways resulting in greater spatiotemporal diversity in cellular signalling [243]. PAC1R activation at the plasma membrane activates the cAMP/PKA and PKC signaling cascades to engage MAP kinase/ERK for neuroplasticity responses [219,244]. In pancreatic β-cells, PAC1R induces transient activation of ERK in both the nuclear and cytosolic compartments via plasma membrane signaling as well as sustained ERK activation in the cytosol, mediated through β-arrestin1 dependent receptor trafficking as described above [219,244].

#### 2.2.6. Receptor Activity-Modifying Proteins (RAMPs)

RAMPs, consisting of RAMP1, 2, and 3, are a family of single transmembrane spanning accessory proteins that associate with many GPCRs and have the potential to modulate their function [245]. As a result of this interaction, RAMPs can act as chaperones to enhance the expression of receptors at the cell surface, alter agonist selectivity of some receptors, influence G protein coupling, and control the trafficking of receptor to endosomal compartments and recycling and degradation pathways [246,247,248].

Using a suspension bead array assay the PAC1R was shown to associate with all three RAMPs [249], albeit the functional consequences, if any, of these interactions are currently unknown. VPAC1R and VPAC2R can also interact with each member of the RAMP family, translocating all three RAMPS to the cell surface (Figure 1E) [168,250]. The physiological relevance of VPAC receptor-RAMP oligomerization remains to be determined; however, functional differences have been observed in VPACR signaling in recombinant systems [164]. The interaction between VPAC1R and RAMP2 enhanced phosphoinositide turnover compared to VPAC1R alone in response to VIP with no change in cAMP stimulation [168]. For VPAC2R, association with RAMP1 and RAMP2 enhanced basal G_i/o_ coupling [250]. However, it must be noted that the VPACR RAMP association can be influenced by cell background as VPAC2R did not promote RAMP trafficking to the cell surface when studied in CHO cells, but was able to promote RAMP trafficking in HEK293 cells [168]. As RAMPs are broadly, but cell specifically expressed in peripheral tissues and in the nervous system and engage with GPCRs differentially in a cell line background-dependent manner [248,251,252], hence investigating VPACR–RAMP complexes in native tissue to observe properties of ligand specificity, G protein coupling, and receptor desensitization and internalization is required to understand the physiological relevance of RAMPs for modulating the function of the PACAP receptor family.

### 2.3. Receptor Desensitisation and Recycling

Ligand-activated receptors can induce signaling from various intracellular compartments [253,254,255]. Endocytosis of the PACAP subfamily of receptors is initiated by the phosphorylation of serine/threonine residues in the intracellular and C-terminal receptor regions [256,257,258,259]. These serine/threonine residues serve as potential phosphorylation sites for protein kinase A (PKA), PKC, and GPCR kinases (GRKs). There are seven GRKs, of which four (GRK2, 3, 5, and 6) are ubiquitously expressed [260]. Phosphorylation by GRKs decouples G protein interaction and thus terminates initial G protein signaling. Phosphorylation of the receptor increases its affinity for β-arrestin recruitment which further facilitates uncoupling of the GPCR from G proteins and can mediate receptor endocytosis as mentioned above [224,261].

PAC1R desensitization and termination of PKC signaling are mediated by GRK3 in human retinoblastoma cells [262], whereas in gastric smooth muscle cells VPAC2R desensitization is mediated by GRK2 [215,263,264]. GRK2, 3, 5, and 6 are all recruited by VPAC1R in HEK293 cells with overexpression of these GRKs, attenuating cAMP production [265]. VPAC1R phosphorylation induces β-arrestin recruitment, however, studies in recombinant cells revealed that overexpression of β-arrestin induced only a minor decrease in cAMP production, suggesting that the mechanism of VPAC1R desensitization may be β-arrestin-independent or VPAC1R may continue to undergo cAMP signaling following β-arrestin recruitment [265,266]. Both VPAC1R and VPAC2R are internalized rapidly following agonist exposure, however, these receptors exhibit distinct trafficking profiles in recombinant systems; VPAC2R is recycled to the cell surface within two hours of receptor internalization, while VPAC1R is not [257,267].

GRK2/3 facilitates β-arrestin1 and β-arrestin2 recruitment to the PAC1R, with different functional consequences. β-arrestin2 recruitment mediates PAC1R internalization and ERK phosphorylation, while PAC1R-β-arrestin1 complexes remain localized at the plasma membrane [268]. In that same study using cortical neuronal cells, silencing of β-arrestin1 increased PAC1R-mediated ERK phosphorylation [268]. However, β-arrestin1 was required for long-lasting ERK1/2 activation by PAC1R in pancreatic islets and for ERK1/2 activation by the PAC1R hop splice variants in HEK293T cells suggesting that the effect of β-arrestin1 on signalling of the PAC1R is cell type-dependent [219,269].

### 2.4. Understanding of PACAP and VIP Signaling and Regulation That Needs to Be Addressed for Disease-Focused Therapies

As described above, the PACAP subfamily of receptors can activate diverse signaling pathways that lead to their pleiotropic effects throughout the body. However, the link between signaling and regulatory mechanisms and the physiological consequences of these mechanisms remain poorly understood. These receptors remain attractive targets for inflammatory and immune diseases, and disorders in the nervous system. However, there are currently no drugs on the market that specifically target PACAP/VIP receptors. While PACAP has been implicated in migraine, current efforts to antagonize PACAP-PAC1R signaling through the development of monoclonal antibodies to PACAP38 or PAC1R for treatment-resistant migraines [85,270,271,272] have not been successful. In phase II clinical trials, the PAC1R monoclonal antibody, AMG301 failed to exhibit efficacy for migraine treatment, despite promising results in preclinical studies [273]. While multiple factors could contribute to this clinical outcome, such as failure to reach the target site of action and inability to block specific signaling pathways associated with the disease, the pharmacological complexities described above including the existence of multiple PAC1R splice isoforms, non-selective PACAP-VPAC receptor signaling and endosomal signaling, present challenges for developing novel therapies. Furthermore, there is increasing evidence that, in addition to forming hetero-oligomers with RAMPs, the PACAP subfamily of receptors can form dimers with themselves and other receptors [172,274,275]. The higher order oligomerization of these receptors may have significant effects on their function. If validated in vivo, this could provide novel therapeutic design strategies. Overall, a lack of understanding of the importance of these factors may impact the therapeutic outcomes of drugs targeting this receptor family. A more detailed investigation into these complexities, including cell and disease-specific expression of receptor splice variants and subtypes, oligomeric complexes, and key signal transduction/regulatory proteins will likely be required for successful clinical translation [168,207,213,214,250,276,277].

Beyond the intricacies of structural variations and potential higher order oligomeric complexes, the ability of the PACAP subfamily to activate multiple downstream signaling pathways enables the potential identification of “biased agonists” that selectively control signaling and thus initiate specific cellular and physiological outcomes. Biased agonism refers to the ability of individual ligands acting at the same receptor to produce distinct profiles of downstream signaling [166,278]. This is of particular therapeutic interest as it means different ligands binding to the same GPCR may preferentially activate beneficial pathways while minimizing those that may result in unwanted side effects [279]. Biased agonism has been observed at various Class B GPCRs, the most extensively studied being the GLP-1R. Through coupling to distinct G proteins (G_s_, G_q_, G_i_._o_) and to β-arrestins, GLP-1R is able to activate a plethora of important signaling pathways in pancreatic β-cells including cAMP production, ERK1/2 phosphorylation, and intracellular Ca^2+^ mobilization [280,281]. One of the earliest reports of biased agonism was at the PAC1R, where both the PACAP27 and PACAP38 stimulated cAMP production in the PAC1nR receptor isoform, while only PACAP38 stimulated IP_3_ production [178]. However, later studies have not necessarily replicated these findings, highlighting the complexities in PACAP subfamily signaling that may influence pharmacological outcomes and mask biased agonism [181,182].

Moreover, pharmacological probes, such as PACAP and VIP, used to study the PACAP/VIP receptors often exhibit some degree of affinity for multiple members within the subfamily. The heterogeneous distribution of these receptors in the body also complicates the attribution of specific receptors of the subfamily to a pharmacological effect, and consequently the physiological outcomes. For example, receptor cross-talk can mask the presence of receptor synergism or antagonism that may potentially occur. Hence, in addition to the therapeutic benefit of next-generation receptor-selective ligands, there is also the potential gain from developing more specific probes for future pharmacological studies.

## 3. Molecular Activation of PACAP and VIP Receptors

A better molecular understanding of the mechanisms behind ligand recognition and receptor activation of the PACAP receptor subfamily can guide the development of more effective clinical candidates and probes. GPCRs share the structural composition of an N-terminal extracellular domain (ECD), seven transmembrane (TM) helices embedded in the lipid bilayer membrane, three extracellular and three intracellular loops connecting the TM helices, and a C-terminal intracellular domain. The class B1 GPCR subfamily, including the VPAC1R, VPAC2R, and PAC1R, are characterized by an N-terminal ECD consisting of approximately 120 amino acid residues and three conserved disulphide bonds that form the core of the ECD structure [282]. The ECD is important for peptide recognition and binding to position the peptide optimally for interaction with the TM core, and subsequent receptor activation [135].

The recent advances in structural biology, particularly the emergence of cryo-electron microscopy (cryo-EM) for GPCR structure determination, have improved our molecular understanding of ligand-occupied GPCRs coupled with their canonical G protein transducer [282]. Breakthroughs in protein complex stabilization strategies, including the use of nanobodies, single-chain fragment antibodies and G protein modifications, such as dominant-negative mutation (DN), mini Gα proteins, and nanoBiT tethering have facilitated the formation and purification of receptor-G protein complexes suitable for structure determination [136,283,284,285,286]. This has been complemented by significant improvements in cryo-EM grid preparation, microscopy, and detector hardware, imaging strategies, and data processing for the three-dimensional visualization of GPCR complexes [282,287,288,289,290,291] and non-GPCRs [292,293].

Before the determination of the full-length PACAP38-bound PAC1R–Gs structure by cryo-EM, there had been some uncertainty on the binding mode of peptides to the N-terminal ECD of the PACAP subfamily of receptors [135]. The crystal structures of the isolated ECD of PAC1sR (PDB: 3N94) and VPAC2R (PDB: 2X57) (Table 5) demonstrated a similar ECD configuration to the crystal structures of other class B1 GPCRs, including the glucose-dependent insulinotropic polypeptide receptor (GIPR) PDB: 2QKH [294] and GLP-1R PDB: 3C5T [295]).

In these structures, three conserved disulphide bonds facilitate folding of the ECD in a sandwich-like configuration consisting of an N-terminal α-helix, two antiparallel β-sheets (β1–β2) and (β3–β4), and a C-terminal α-helix. The first disulphide bond links the N-terminal α-helix and β1–β2 sheet, the second the two β-sheets while the final bond between C77^ECD^ and C113^ECD^ connects the β3–β4 sheet to the C-terminal helix. The peptide-binding clefts in these ECD structures resides between the two central anti-parallel β-sheets, consistent with other class B1 GPCRs ECD structures [297]. In contrast, the NMR structure of PACAP(6–38) complexed to PAC1sR (PDB: 2JOD) exhibited differences to the crystal structure in the topology of the area between β3 and β4 and a distinct peptide-binding mode compared to other class B1 GPCRs [298]. While the first full-length active cryo-EM structure of the PACAP receptor subfamily, the PACAP38-bound PAC1R complex (PDB: 6P9Y), had limited resolution in the ECD, this structure, and subsequently structures of VPAC1R and VPAC2R, confirmed that the peptide binding mode of the PACAP/VIP receptors is similar to other class B1 GPCRs (Figure 3), and the binding mode observed in the crystal structure of the PAC1R ECD [135].

### 3.1. Structural Characteristics of the PACAP Subfamily of Receptors

Structures of PAC1, VPAC1, and VPAC2 receptors determined by cryo-EM are consistent with the proposed class B1 GPCR two-domain activation model, where the C-terminal portion of the peptide agonist binds to the peptide-binding cleft within the ECD [135,299], while the N-terminal helix of the peptide interacts with the receptor core. The N-terminal ECD of the receptor forms a horseshoe-like configuration around the peptide, extending towards the receptor core to form interactions with the top of ECL1 [135,136,137,300,301,302,303,304,305,306,307,308,309].

**Figure 3 ijms-23-08069-f003:**
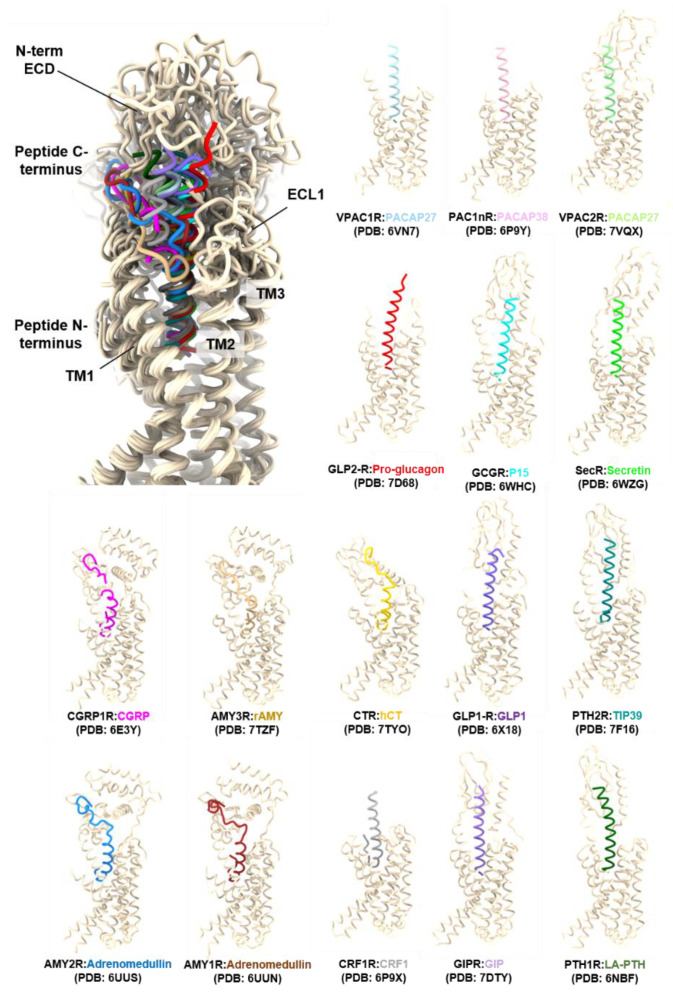
Comparisons of the binding mode of representative peptides bound to Class B1 GPCRs reveal a shared peptide binding mode where the N-terminus of the peptide interacts with the transmembrane (TM) core and the C-terminus of the peptide interacts with the N-terminal extracellular domain (ECD) and extracellular loops (ECLs) of the receptor [135,136,137,300,301,302,303,304,305,306,307,308,309]. The highest variability of the structures is in the extracellular domains and peptide C-termini (as displayed in the structure overlay in the first panel. Structures were aligned by receptor chains and displayed as ribbons (licorice style) using Chimera X 1.3 [310,311]. G protein subunits are not displayed for clarity. The following models lack extracellular domains due to low resolution: PDB 6VN7, 6P9Y, 7D68, and 6P9X.

Due to the greater flexibility and mobility of the ECD regions, ICLs and ECLs, these regions remain relatively less well-resolved compared to the TM domains. Details of the agonist, global resolution, and receptor and G protein modifications of each of the structures of the PACAP subfamily of receptors determined by cryo-EM are listed below (Table 6). In addition to the differing resolutions achieved in the structures, some of the domains/loops are missing in the PDB models due to the limited EM-map resolution in these regions. Namely, for the PACAP27-bound VPAC2R (PDB: 6VN7) and PACAP38-bound PAC1nR (PDB: 6P9Y) complexed with DNGα_s_ structures, the N-terminal ECD, ECL1, and ICL3 are not modeled in the final deposited PDB file. While for the PACAP38-bound PAC1nR complexed with mini-Gα_s_ structure (PDB: 6LPB), ICL1, ECL1 and ICL3 are absent. For the PACAP38- (PDB: 6M1I) and maxadilan-bound (PDB: 6M1H) PAC1sR and PAC27-bound VPAC2R (7VQX) structures, ICL3 is not modeled in the PDB files.

Thus far, the published cryo-EM structures have been solved with either PACAP isoforms or maxadilan. Hence the binding characteristics of these are described below.

### 3.2. Characteristics of the PACAP-Bound VPAC1R, VPAC2R, and PAC1R Structures

Of the six structures determined by cryo-EM to date, five were PACAP-bound. The three PACAP-PAC1R structures (PDB: 6P9Y, 6LPB, and 6M1I) share an agreement on the PACAP binding mode (Figure 4A,B). Of these structures, two are of the receptor splice variant with the full-length ECD, PAC1nR [131,135], while the other is of PAC1sR, missing twenty residues (Val89^ECD^–Ser109^ECD^) in the N-terminal ECD [134]. While this region of the ECD is critical for the binding affinity of VIP, with an almost hundred-fold improvement in binding affinity observed for PAC1sR compared to PAC1nR, the two receptor variants share similar PACAP binding affinities [175]. The additional residues in the ECD that are present in the PAC1nR splice variant have not been resolved in any of the cryo-EM structures to date, most likely due to high flexibility.

Comparison between the PACAP-bound PAC1R, VPAC1R, and VPAC2R structures indicates that the N-terminus of the peptide is inserted into the receptor core at a similar orientation and angle for each receptor. By contrast, greater differences are observed in the C-terminal half of PACAP interacting with the ECL1, the extracellular tip of TM1, and the ECD (Figure 4A,B).

While the PAC1R structures were determined bound to PACAP38, the VPAC1R and VPAC2R were bound to PACAP27. However, due to the high mobility of the PACAP C-terminus in PACAP38-bound structures, only the first 27 residues were modeled. When comparing these structures, the interactions of the first three residues of the relevant PACAP with the receptor core of the VPAC1R, VPAC2R, and PAC1R (Figure 5A–C), from studies of alanine substituted peptides, the N-terminal His^1P^ (superscript is the residue number of the peptide) and Asp^3P^ residues are crucial for binding affinity and biological activity of the peptide for all three receptor subtypes [142,312,313]. This is reflected in the extensive, conserved, hydrophobic contacts between His^1P^ and the receptor TM3 and TM5 and the conserved salt bridge between Asp^3P^ and Arg^2.60^ (the superscript number refers to the Wootten et al. [314], class B numbering system) that is conserved across all receptors of the PACAP subfamily (Figure 5). At the N-terminal end of PACAP, His^1P^ forms extensive hydrophobic interactions in the TM bundle with TM3 (Val^3.40^, Val^3.13^) and TM5 (Trp^5.36^) of all three receptors. However, subtle differences in interactions are observed where His^1P^ also forms a hydrogen bond with Gln^3.37^ of TM3 in VPAC2R, while no hydrogen bond was observed for the equivalent His^3.37^ residue in PAC1R (Figure 5B). Asp^3P^ forms a salt bridge with Arg^2.60^ and hydrophobic contacts with TM7 (Leu^7.43^) that are shared in all three receptors (Figure 5C). The residues at position 1 and position 3 in PACAP, His^1P^, and Asp^3P^, respectively are highly conserved in class B1 GPCR peptides. On the other hand, at position 2, Ser^2P^ of PACAP forms a hydrogen bond with Tyr^3.44^ lower down (from the extracellular face of the receptor) on PAC1R TM3 resulting in a deeper insertion of the peptide into the PAC1R core than observed with the VPACRs (Figure 5B). For VPAC2R, Ser^2P^ forms a hydrogen bond with TM7 (Glu^7.42^) that is not observed in VPAC1R nor PAC1R (Figure 5B).

The PACAP residues Gly^4P^, Ile^5P,^ and Phe^6P^ make extensive interactions with the hydrophobic pocket residues of TM1 and TM7 (Figure 5D). Many of these interactions are conserved between the three receptors. Specifically, Ile^5P^ makes hydrophobic contacts with the equivalent residue at 7.39 in all three receptors, while the aromatic residue of Phe^6P^ interacts with Tyr^1.36^, Val^1.39^ and Tyr^1.43^ residues in TM1 and Leu^7.43^ in TM7 (Figure 5D). Phe^6P^ of PACAP is crucial for binding affinity and biological activity at all three PACAP receptor subtypes as reflected by the loss of biological activity with alanine substitution [142,312,313,315]. However, the replacement of Phe^6P^ with bulky hydrophobic groups modulates the specificity of PACAP analogues for different receptors to some extent [312,313,316,317,318]. For example, the substitution of Phe^6P^ in PACAP with an amino acid with two aromatic rings in the case of [4,4′-biphenylalanine^6^]-PACAP27 resulted in an approximately 10-fold reduction in VPAC1R -mediated Ca^2+^ mobilization potency compared to PACAP27, whilst retaining similar potency at the PAC1R [312]. On the other hand, substitution with the two aromatic ring amino acids, 1-naphthylalanine^6P^ resulted in a 3-fold reduction in potency in Ca^2+^ mobilization at the PAC1R and VPAC1R, and a 10-fold reduction in potency at the VPAC2R, compared with PACAP27 [312]. Gly^4P^ is in closer proximity to Trp^5.36^ of TM5 in PAC1R compared to the VPACRs (Figure 5D). In VIP, the equivalent residue at position 4 is an alanine and at position 5 is a valine. Due to steric hindrance, the additional methyl group in alanine and the different steric profile of the branched alkyl side of valine compared with glycine and isoleucine, respectively, may restrict the binding of VIP to PAC1R, which may contribute to the selectivity of VIP for the VPAC receptors [318,319]. The substitution of PACAP residues Gly^4P^ and Ile^5P^ with Ala^4P^ and Val^5P^ of VIP reduced PACAP affinity for PAC1R when tested in guinea pigs and the rat brain [320,321], suggesting these residues are important for receptor selectivity.

In the cryo-EM structures, high-resolution features are observed for the TM regions, suggesting the peptide bound-TM core is relatively stable, hence side-chain rotamers could be accurately modeled [135,136,137,138,139]. In contrast, the ECLs and ECD are relatively mobile, hence these regions are not as well resolved in the cryo-EM maps, limiting the accuracy of the modeling for side chain rotamers in these regions. Nonetheless, ECL2 is better resolved than ECL1 and ECL3 in many of these structures. Ser^11P^ contributes to polar interactions with ECL2 (with Asp287^ECL2^, Asp273^ECL2,^ and Asp298^ECL2^ of VPAC1R, VPAC2R, and PAC1R, respectively) for all three receptors (Figure 5E). In contrast, the interaction between Asp^8P^ of PACAP with the ECL2 residues Ile289^ECL2^ and Asn275^ECL2^ of VPAC1R is not observed in PAC1R. On the other hand, Lys^2.67^ in VPAC2R forms a hydrogen bond with Thr^7P^ in VPAC2R, while this interaction is not observed in PAC1R and VPAC1R (Figure 5E) [135,136,137].

The C-terminal end of the peptide of the VPAC2-bound PACAP27 was shifted by 4.6Å toward the TMD core compared to VPAC1R-bound PACAP27 when comparing the α-carbon atom of Leu^27P^ (Figure 5F). In comparison, the C terminus of PACAP27-bound VPAC2R moved towards ECL1 by 3.1 Å when compared with residue Leu^27P^ of the PACAP38-bound PAC1nR (PDB: 6P9Y) (Figure 5G) [135,136,137]. The PACAP38-bound VPAC1R (PDB: 6VN7) and PACAP38-bound PAC1nR (PDB: 6P9Y) structures did not include modeling of ECL1 and the ECD due to limited resolution in these regions, however, Xu, et al., 2022 reported that the N-terminal α-helix of the VPAC2R ECD adopts a unique conformation, relative to other class B1 GPCRs, and compared to the modeled ECD in the PACAP38-bound PAC1sR (PDB: 6M1I). In VPAC2R, the N-terminal α-helix inserts deeply into a cleft between PACAP27 and the predicted, more open, position of ECL1 in the model, stabilizing the peptide-receptor interface (Figure 5G) [137]. Nonetheless, the limited resolution in the N-terminal ECD and ECLs and the accuracy of modeling rotamers in this region must be taken into consideration when interpreting interactions between the peptide and the receptor ECD and ECL1.

### 3.3. Characteristics of the Maxadilan-Bound PAC1R Structure

The PAC1R-selective agonist maxadilan is a 61-amino acid peptide that consists of two α-helices (helix 1 and helix 2) that form a v-shaped conformation held together by a disulphide bond between Cys^14P^–Cys^51P^ [133,138,322,323]. Maxadilan binds to the same binding pocket as PACAP on PAC1R (Figure 4C). However, to accommodate the larger 3D conformation of maxadilan, TM1, 6 and 7 of PAC1R are shifted outward compared to the PACAP38-bound structures (Figure 4C,D) [138]. Helix 1 of maxadilan interacts with TM1 of PAC1sR on one side of the receptor, with the residues Ser^21P^ and Gln^25P^ of helix 1 of maxadilan forming hydrogen bonds with Asp147^1.33^ and Tyr150^1.36^ of TM1. On the other side of the receptor binding pocket, the loop of maxadilan and helix 2 forms hydrogen bonds with TM3 (Ser^33P^ with Tyr241^3.44^) and ECL2 (Thr^39P^-Asp301^ECL2^) (Figure 4E). At the receptor core, the loop of maxadilan is reported to form hydrophobic interactions with TM6 (Phe^34P^ with Phe369^6.56^ and Ala370^6.57^) and TM7 (Ala^32P^ with Leu386^7.43^) [138].

Truncated forms of maxadilan, max.D.4 (missing amino acids 24–42) and M65 (missing amino acids 25–41) act as PAC1R-specific antagonists [133,148,324]. Both max.D.4 and M65 have residues in helix 1 and the loop of maxadilan removed. These regions of maxadilan interact with the binding site in the maxadilan-bound PAC1sR structure, suggesting their importance for agonist activity. Namely, the hydrogen bonds formed between helix 1 of maxadilan and TM1 (Gln^25P^-Tyr150^1.36^) and between the loop of maxadilan and ECL2 (Ser^33P^-Tyr241^3.44^ and Thr^39P^-Asp301^ECL2^) as well as hydrophobic contacts between Ala^32P^ and Leu386^7.43^ of TM7 and the amide-Pi stack interactions between Phe^34P^ and TM6 (Phe369^6.56^ and Ala370^6.57^) would be predicted to be lost for the antagonist maxadilan analogues [138].

### 3.4. Activation Characteristics of the PACAP Subfamily of Receptors

While the PACAP receptors can potentially couple to G_q/11_ and G_i/o_ proteins and recruit β-arrestins for the activation of downstream signaling [214,215,268], currently all active structures of the PACAP receptor subfamily are in complex with G_s_. Hence, the elucidation of structures of these receptors with non-G_s_ transducer proteins will be important in understanding the molecular mechanisms behind the diversity in transducer signaling. The G_s_-coupled, active structures of the PACAP receptor subfamily members display similar characteristics to other active class B1 GPCRs [135,282,299,325]. As there are currently no inactive state structures of PACAP subfamily receptors, activation mechanisms are inferred using the inactive structures of other class B GPCRs and are generally limited to global activation mechanisms observed for class B1 GPCRs.

Deep engagement of peptide agonists with the TM bundle of class B1 GPCRs above a central polar network is proposed to promote receptor activation, with two additional layers of conserved polar networks within the TM bundle involved in this process. For the PACAP family receptors, the central conserved polar network (top layer) is formed by Arg^2.60^, Asn^3.43^, His^6.52^ and Gln^7.49^ [282,314,325,326,327], the middle being the HETx polar network consisting of residues His^2.50^, Glu^3.50^, Thr^6.42^ and Tyr^7.57^ [282,325], and the bottom at the cytoplasmic face, consisting of the TM2-6-7-8 network formed by Arg^2.46^, Arg^6.37^, Asn^7.61^, Glu^8.41^ [282,325] (Figure 6).

The binding of peptide agonists leads to reorganization of the ECLs and outward movement of the TM6/ECL3/TM7 and inward movement of TM1 (Figure 6A) [282,325,328,329,330,331]. The peptide and G protein binding pockets are allosterically connected, hence both facilitate the process of receptor activation [332]. Interactions in the central polar network are reorganized by peptide and G protein engagement, inducing the unraveling of TM6 around the Pro^6.47^xxGly^6.50^ motif, resulting in TM6 undergoing an approximate 90° kink [325,333]. Gln^7.49^ from the central polar network reorganizes to interact with Tyr^6^.^53^, and Asn^5.50^ forms hydrogen bonds with Leu^3.47^, while residues Phe^5.54^, Ile^5.57^ and Ile^5.58^ form extensive hydrophobic contacts with the main chains of TM3 and TM6 that stabilize the active conformation [136,282,334]. The activated conformation(s) is further stabilized through coordinated water networks across the TM domain [300,335,336].

The outward movement of the intracellular half of TM6 and a bend of the extracellular portion of TM7 towards TM6 results in rearrangement in the lower polar networks (Figure 6B) [282,325]. The destabilization of the HETx motif (His^2.50^, Glu^3.50^, Thr^6.42^ and Tyr^7.57^) breaks ground state interactions between Thr^6.42^ and Tyr^7.57^, and the Arg^6.40^–Glu^8.41^ interaction is lost [282]. Intracellular hydrogen bonds in the TM2-6-7-helix 8 polar network that are present in inactive state class B1 GPCR structures are broken in the peptide and G protein bound PAC1R, VPAC1R and VPA2R structures, enabling the insertion of the C-terminal α5-helix of the Gαs subunit into the core of the receptors, where the α5 helix of G_s_ forms conserved interactions with TM2, TM3, and ICL2 enabling G protein activation [282,325].

### 3.5. Allosteric Modulation

Due to the high degree of conservation in the TM bundle of different PACAP subfamily receptors where the peptide agonist activation domains bind, the identification of allosteric ligands may be an additional strategy that could be used for developing more selective therapeutics [337]. Allosteric ligands bind to a spatially distinct binding site than the endogenous agonists and therefore have the potential to modulate the receptor response to these agonists. Since allosteric sites are likely to be less evolutionarily conserved compared to the natural orthosteric sites, targeting an allosteric site could engender receptor selectivity or alter the signaling/regulatory profile of peptide-bound receptors [338].

Allosteric ligands resulting in positive effects on the affinity and/or efficacy of the orthosteric ligand response are known as positive allosteric modulators (PAMs), while allosteric ligands resulting in negative effects on the orthosteric ligands are known as negative allosteric modulators (NAMs) [338]. Only limited studies have attempted to discover allosteric ligands within the PACAP subfamily of receptors, however, this may be an avenue to explore for future drug design.

From in silico screening of the ZINC15 drug library, ticagrelor was proposed as a potential allosteric modulator for the PACAP subfamily of receptors. Ticagrelor, an approved antiplatelet medicine, was proposed as a NAM for inhibiting VIP-induced calcium mobilization in CHO cells expressing either VPAC1R or VPAC2R, with weak selectivity towards the VPAC2R [339]. Using molecular dynamics simulations with an inactive homology model of VPAC1R that was generated based on the GLP1-R crystal structure (PDB: 5VEW) [340,341], the allosteric ticagrelor binding site was predicted to be located in the intracellular region of the TM bundle, in a pocket formed by Arg^6.37^, Arg^6.40^, Arg^2.46^ and Asn^8.47^, and therefore proposed to prevent conformational changes in the region of the tyrosine Thr^6.42^-Tyr^7.57^ toggle switch required for VPAC1 and VPAC2 receptor activation [339]. The TM helical bundle of class B1 GPCRs is a region that has attracted much attention in allosteric drug design as modulation of the TMs, in particular TM6, could be targeted to stabilize or restrict an active conformational receptor state [337]. NNC0640 and MK-0893 are NAMs that bind to TM6 in the inactive glucagon receptor (GCGR) crystal structures (PDB: 5XEZ and 5EE7 respectively), restricting the outward movement of the TM6 intracellular domain [329,342].

Positive allosteric modulators of the PAC1R have also been reported with doxycycline and minocycline proposed to enhance PACAP binding to PAC1R by binding to a site in the N-terminal ECD [343]. Doxycycline and minocycline belong to the antimicrobial class of tetracyclines and promote the expression of the plasminogen activators: tissue plasminogen activator (tPA) and urokinase plasminogen activator (uPA) via PAC1R activation in RT4 Schwann cells and augment PACAP27-induced cAMP stimulation in CHO cells overexpressing PAC1R [344]. While the 3D configuration of the N-terminal ECD is conserved across class B1 GPCRs and within the PACAP receptor subfamily, the amino acid composition is relatively diverse [337]. Hence, targeting regions of the ECD provides opportunities for the development of subtype-selective PACAP receptor allosteric modulators. This may be particularly true for VPAC2R, where the PACAP27-bound VPAC2R Gs structure (PDB: 7VQX) demonstrated that the N-terminal α-helix of the ECD adopts a unique conformation by inserting into a cleft between the peptide and ECL1 to stabilize the peptide-receptor interface [137].

## 4. Concluding Remarks

Historically, the predominant strategy for the discovery of clinical candidates for the PACAP receptor subfamily has been based on structural modifications to endogenous peptides. Over the last decade, there has been considerable progress in the determination of the three-dimensional structures of these receptors using cryo-EM, in complex with agonists and G proteins, for the identification of key regions for peptide recognition and selectivity, which complements earlier X-ray crystallography on receptor subdomains. These recent structures advance our understanding of previously reported peptide and receptor structural-activity relationships from empirical drug development and provide insight into key regions of the peptide binding site to enable strategies for rational, structure-guided peptide design.

While the ECD, where the peptide C-terminus initially recognizes class B1 GPCRs, remains less well-resolved than other regions, the resolution of structures determined through cryo-EM has drastically improved over the past five years through advances in sample preparation, data collection, and data processing, and is predicted to continue improving with sub-2.5 Å cryo-EM structure determination being increasingly routine for class B1 GPCR–G protein complexes [288]. Moreover, direct information on 3D conformational dynamics captured during vitrification can increasingly be extracted through sophisticated analytical methods [304,345] and such dynamics play critical roles in peptide recognition and modes of receptor activation that may not be apparent from high-resolution consensus reconstructions [304,332,346,347]. This information can be integrated with other methods for interrogating GPCR dynamics, including hydrogen deuterium exchange mass spectrometry (HDX-MS) [348], single-molecule FRET studies [349] and electron paramagnetic resonance (EPR) [350,351] and holds promise for further enhancing peptide drug design [348,352]. This emerging knowledge on the different conformational states and structural features responsible for activation transition from inactive and apo-state GPCRs could be utilized in guiding the design of peptides of distinct pharmacology and selectivity for different receptors within the PACAP subfamily as well as identifying allosteric binding sites for new compounds [348,353].

The next question is: how do we use this information to develop better peptide drugs and selectively target signaling pathways activated by PACAP receptors for specific physiological outcomes? As described above, the PACAP subfamily of receptors can activate diverse signaling pathways leading to their pleiotropic effects throughout the body and are attractive targets for inflammatory and immune diseases, and disorders in the nervous system. To coordinate the diverse physiological outcomes mediated by the activation of the PACAP subfamily of receptors, these receptors activate multiple signaling pathways through the coupling of different G proteins (e.g., G_s_, G_q/11,_ and G_i/o_ proteins) as well as through associating with β-arrestin and RAMPs to modulate receptor signaling and trafficking [168,178,192,207,250,268]. Additionally, splice variants within the PAC1R, as well as the potential for biased agonism through these receptor variants, may contribute to differential transducer coupling and downstream signaling, encoding for functional selectivity [178,180].

Biased peptide agonists, which have been broadly observed across class B1 GPCRs, have therapeutic potential through coupling distinct profiles of transducers that produce beneficial clinical effects, while reducing undesirable side effects [166,280,354]. All PACAP subfamily receptor structures determined to date have been Gα_s_ protein-coupled with modifications that stabilize the receptor G protein complex. Hence future structural studies with non-G_s_ transducers will determine whether signaling bias can be structurally explained for the rational design of biased peptides for PACAP receptor subtypes. Furthermore, short half-lives, lack of oral bioavailability, and difficulty in penetrating the blood–brain barrier are challenges observed with many peptides. As many of the therapeutic targets of the PACAP receptor subfamily are located in the CNS, the engineering of peptides that can selectively bind to the PACAP receptors and cross the blood–brain barrier remains a major challenge.

## Figures and Tables

**Figure 1 ijms-23-08069-f001:**
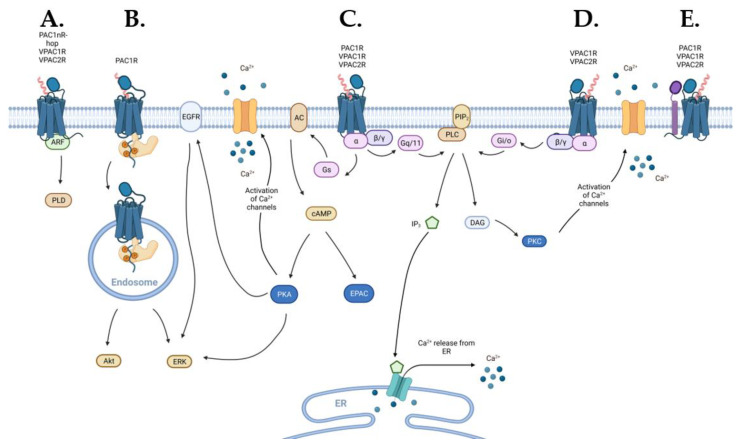
Schematic diagram of the signaling pathways activated by the PACAP subfamily of receptors (blue): (**A**) VPACRs and the PAC1R splice variant PAC1nR-hop can engage ADP-ribosylation factor (ARF) signaling, (**B**) endosomal signaling may be activated by β-arrestin (light-orange) recruitment to PAC1R, (**C**) VPAC1R, VPAC2R, and PAC1R directly engage G_s_- and G_q/11_-coupled pathways, (**D**) VPAC1R and VPAC2R also directly activate G_i/o_-coupled pathways, and (**E**) PACAP subfamily of receptors can interact, in a receptor-specific manner, with receptor activity-modifying protein (RAMP) (purple) that can alter receptor signaling or trafficking. Abbreviations used in this figure: ARF—ADP-ribosylation factor, PLD—phospholipase D, ERK—extracellular signal-regulated kinase, EGFR—epidermal growth factor receptor, AC—adenylate cyclase, cAMP—3′,5′-cyclic adenosine monophosphate, PKA—protein kinase A, EPAC—exchange protein directly activated by cAMP, Ca^2+^—calcium, PLC—phospholipase C, PIP_2_—phosphatidylinositol 4,5-bisphosphate, IP_3_—inositol 1,4,5-trisphosphate, DAG—diacylglycerol, ER—endoplasmic reticulum, PKC—protein kinase C.

**Figure 2 ijms-23-08069-f002:**
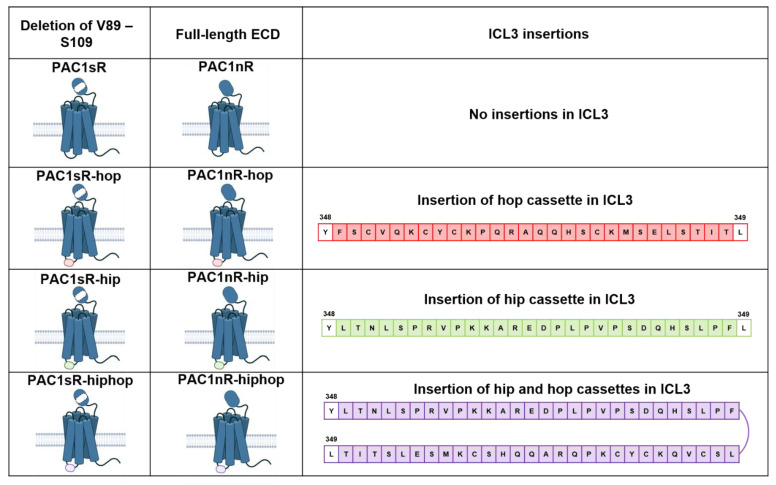
Schematic illustration of PAC1R splice isoforms occurring in the N-terminal extracellular domain (ECD) and/or intracellular loop 3 (ICL3) in humans. N-terminal ECD splice variation arises from the presence (PAC1R null a.k.a. PAC1nR) or deletion (PAC1R-short a.k.a. PAC1sR) of 21 amino acids (residues V89 to S109) from the N-terminal ECD. ICL3 splice variation arises from the inclusion of one or two of the cassette exons, the “hip” cassette (red) and the “hop” cassette (green) of 28 amino acids each, between residues Y348 and L349 of the PAC1R ICL3. Inclusion of the hip cassette in PAC1R leads to the splice isoform PAC1nR-hip or PAC1sR-hip. Inclusion of the hop cassette leads to the PAC1nR-hop or PAC1sR-hop splice isoform, while the inclusion of both hip and hop cassettes leads to PAC1nR-hiphop or PAC1sR-hiphop (purple).

**Figure 4 ijms-23-08069-f004:**
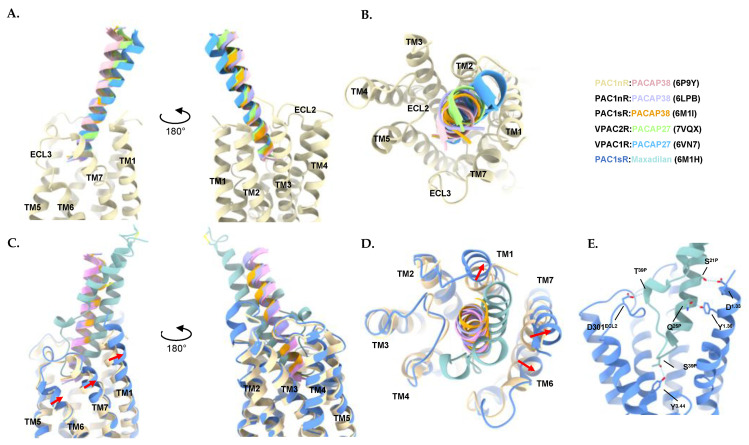
Comparison of peptide binding mode to receptors of the PACAP subfamily. (**A**) Side views and (**B**) Top view of the superimposed structures of PACAP bound to VPAC1R, VPAC2R, and PAC1R. Only the transmembrane (TM) helices of the receptor chain of PAC1nR (PDB: 6P9Y) are shown for reference/clarity. (**C**) Side views and (**D**) Top view of the superimposed structures of maxadilan and PACAP bound to PAC1R reveals TM1, 6, and 7 shifted outward to accommodate maxadilan (PDB: 6M1H) when compared to the PACAP38-bound PAC1nR (PDB: 6P9Y). Movement of the TMs indicated by red arrows. Only the TM helices of the receptor chain of the PACAP38-bound PAC1nR (PDB: 6P9Y) and the TM helices of the receptor chain for the maxadilan-bound PAC1sR (PDB: 6M1H) are shown for reference/clarity. (**E**) Helix 1 of maxadilan forms hydrogen bonds with TM1 in the peptide binding pocket while the loop of maxadilan and helix 2 forms hydrogen bonds with ECL2 and TM3. Structure is shown as a side view with a focus on the receptor core. Hydrogen bonds calculated with ChimeraX 1.3. Structures were aligned by receptor chains and displayed as ribbons, with residues involved in hydrogen bonds (dotted lines) shown as sticks using ChimeraX 1.3 [310,311]. Colors for the peptides and receptors are shown in the figure. Peptide residues are denoted with superscript P, extracellular loop residues are denoted as ECL. Receptor transmembrane residues are denoted using the Wootten numbering system.

**Figure 5 ijms-23-08069-f005:**
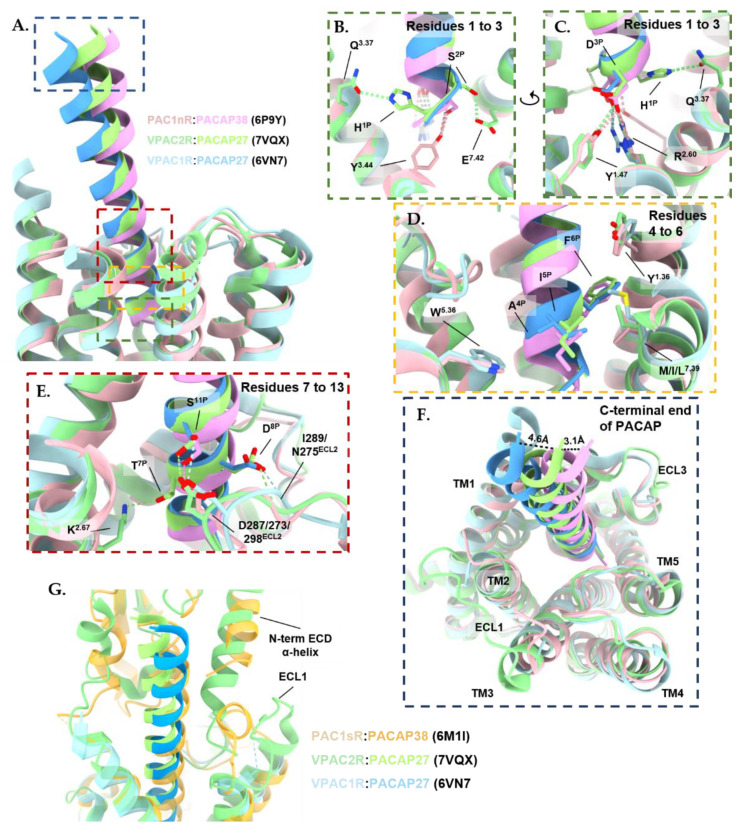
Comparison of PACAP in the binding pocket of VPAC1R (blue; PDB: 6VN7), VPAC2R (green; PDB: 7VQX) and PAC1nR (pink; PDB: 6P9Y). Structures were aligned by receptor chains and displayed as ribbons, with residues involved in hydrogen bonds (dotted lines) and hydrophobic interactions shown as a stick; calculated with ChimeraX 1.3 [310,311]. Extracellular domain (ECD) residues of the receptor were removed for clarity. Peptide residues are denoted with superscript P, extracellular loop residues are denoted as ECL. Receptor transmembrane (TM) residues are denoted using the Wootten numbering system. (**A**) Overview of comparison highlighting the regions of higher magnification displayed in panels (**B**–**E**) with matching colored dashed rectangles. (**B**,**C**) Close-up view of the hydrogen bonds formed between the first three residues of PACAP and the binding pocket of VPAC1R (blue), VPAC2R (green), and PAC1nR (pink). (**D**) Close-up view (front and back side view) of PACAP residues 4 to 6 that interact with conserved hydrophobic side chains in TM1 (Y^1.36^), TM5 (W^5.36^) and TM7 (M/I/L^7.39^) of the peptide binding pocket of VPAC1R (blue), VPAC2R (green) and PAC1nR (pink). (**E**) Close-up view of hydrogen bonds formed between PACAP from residue 7 to 13 and the binding pocket of VPAC1R (blue), VPAC2R (green), and PAC1nR (pink). (**F**) Close-up view of the C-terminal end of PACAP peptide that is shifted 4.6 Å towards the receptor core in VPAC2R compared to VPAC1R, while the C-terminal end of PACAP peptide is shifted 3.1 Å towards ECL3 in PAC1R compared to VPAC2R. Distance between the α-carbon of Leu27P in PACAP peptide between the three structures calculated with ChimeraX 1.3. (**G**) The N-terminal α-helix of the VPAC2R ECD (green; PDB: 7VQX) adopts a unique conformation that deeply inserts into a cleft between PACAP27 and the predicted position of ECL1 in the model compared to the modeled ECD in the PACAP38-bound PAC1sR (yellow; PDB: 6M1I). Due to limited resolution in the N-terminal ECD and ECL1, these regions are not modeled in the PACAP27-bound VPAC1R structure (blue; PDB: 6VN7).

**Figure 6 ijms-23-08069-f006:**
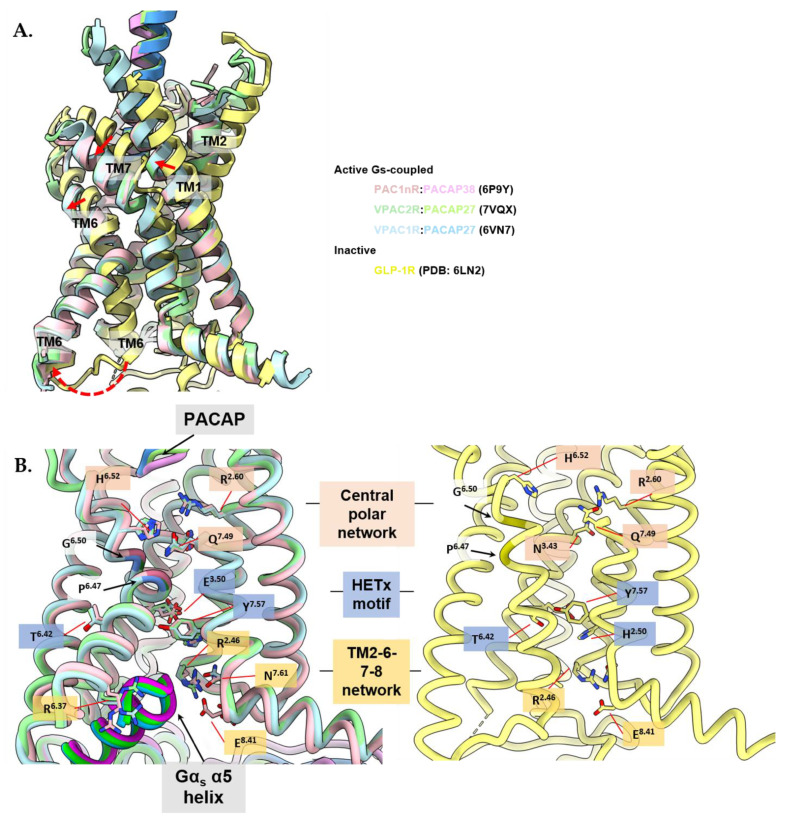
The active receptor conformation is similar between members of the PACAP subfamily of receptors. Active, G_s_-coupled PACAP-bound receptors are displayed: VPAC1R (blue; PDB: 6VN7), VPAC2R (green; PDB: 7VQX), and PAC1nR (pink; PDB: 6P9Y) and compared to the inactive crystal structure of full length human GLP1 receptor (GLP1-R) in complex with Fab fragment (Fab7F38) (yellow; PDB: 6LN2). Structures were aligned by receptor chains and displayed as ribbons as side views, with important residues shown as sticks using ChimeraX 1.3 [310,311]. Gαs-α5 helix colored in darker colors corresponding to the respective receptor chains (**B**, left panel). The Fab fragment in inactive GLP1-R was omitted for clarity. (**A**) Comparison between active structures of the PACAP subfamily of receptors and the full-length inactive structure of GLP1-R indicate inward movement of TM1 and outward movement of TM6 and 7 upon receptor activation with TM6 undergoing a kink at the Pro^6.47^xxGly^6.50^ motif. Movement of TMs indicated by red arrows. The G protein was omitted for clarity. (**B**) Comparison between conserved residues in the central polar network (pink shading-top), HETx motif (blue shading-middle) and TM2-6-7-8 network (yellow shading-bottom) between the active PACAP subfamily of receptor structures (left panel) and the inactive GLP1-R structure (right panel) indicate rearrangement of the conserved polar networks for TM6 to kink at the Pro^6.47^xxGly^6.50^ motif and accommodate the α5 helix of Gαs at the cytoplasmic face.

**Table 1 ijms-23-08069-t001:** Distribution and physiological or therapeutic role of the PACAP subfamily of receptors.

Receptor	Distribution	Physiological/Therapeutic Role
VPAC1R	CNS [35] (cerebral cortex [36,37], hypothalamus [38], hippocampus [36,39])	Control of circadian rhythm [38], learning, and memory [39]
	Liver [40]	Glucose metabolism [41,42]
	Lung [40,43]	Asthma and COPD (relaxation of airway and vascular smooth muscles [44], anti-inflammatory effect [45,46], and regulation of mucus secretion [47]), chronic bronchitis [48]
	Intestine [40,49]	Peristalsis, ion transport and mucus secretion [11,12,50]
	Breast [40]	Cell proliferation in cancer [51,52]
	T-lymphocytes and macrophages (constitutively expressed) [53,54,55,56]	Immune regulation [32,33]
VPAC2R	CNS [35] (thalamus [57,58,59], suprachiasmatic nucleus [29,31,57,59], dentate gyrus [59], amygdala [57])	Schizophrenia [60,61,62], brain injury [63], control of circadian rhythm [29,30,31], processing of fear-related memory [64]
	Smooth muscles [65]	Vasodilation (blood vessels) [66], erectile dysfunction (male reproductive system) [67]
	Pancreas [68]	Insulin secretion [34]
	Lungs [65,69]	Asthma and COPD (relaxation of airway and vascular smooth muscles [44,70], anti-inflammatory effect [45,46], and regulation of mucus secretion [47]), pulmonary arterial hypertension [71], chronic bronchitis [48]
	T-lymphocytes and macrophages (expressed upon cell activation) [53,54,56]	Immune regulation [72]
PAC1R	CNS [35] (olfactory bulb [16,73], cerebral cortex, thalamus [73], hypothalamus [73,74], hippocampus, amygdala [73,75], substantia nigra [73], cerebellum [73]	Astrocyte proliferation [76], appetite and feeding behaviour [77,78], anxiety [79], stress response [80,81,82], control of circadian rhythm [20], post-traumatic stress disorder [83], traumatic brain injury [84], migraine [85], Alzheimer’s disease [86,87]
	Embryonic nervous system [88,89]	Neuronal differentiation of neural progenitor and embryonic stem cells [90,91,92]
	Eyes (corneal endothelium [93], retina [94], lacrimal gland [95]	Maintenance of corneal endothelial barrier integrity [93] Protection against retinopathy [94,96] Stimulation of tear production [95]
	Bone marrow (haematopoietic progenitor cells) [97]	Haematopoiesis [97]
	Adrenal medulla	Adrenal catecholamine secretion [98]
	Pancreas	Insulin secretion [99]
	Cardiac neurons [100]	Modulates excitability—stimulatory effect on CV system [100]
	Bladder	Urinary bladder dysfunction [101]

**Table 2 ijms-23-08069-t002:** VPAC1R, VPAC2R, or PAC1R selective peptide analogues.

Selective Receptor	Compound	Agonist/Antagonist	Peptide Modifications	Relative Selectivity *	Reference
VPAC1R	[Tyr^9^,Dip^18^]-VIP	Agonist	VIP analogue	VPAC1R ([^125^I]VIP K_i_ = 0.1 nM) VPAC2R ([^125^I]VIP K_i_ = 53 nM) PAC1R ([^125^I]PACAP27 K_i_ = 3 μM)	[140]
	[Ala^22^]-VIP	Agonist	VIP analogue	VPAC1R ([^125^I]VIP IC_50_ = 10 nM), VPAC2R ([^125^I]RO 25-1553 IC_50_ = 1 μM)	[141,142]
	[Leu^22^]-VIP	Agonist	VIP analogue	VPAC1R ([^125^I]VIP IC_50_ = 11 nM), VPAC2R ([^125^I]RO 25-1553 IC_50_ = 700 nM)	[143]
	[Ala^11,22,28^]-VIP	Agonist	VIP analogue	VPAC1R (cAMP EC_50_ < 1 nM) VPAC2R (cAMP EC_50_ > 1 µM)	[142]
	[Arg^16^]-PACAP (1–23)	Agonist	C-terminal truncated PACAP analogue	VPAC1R ([^125^I]VIP IC_50_ = 2.5 nM), VPAC2R ([^125^I]RO 25-1553 IC_50_ = 1.2 µM)	[143]
	Chicken [Arg^16^]-secretin	Agonist	Secretin analogue	PAC1R ([^125^I]Ac-His^1^-PACAP27 IC_50_ = 30 µM) VPAC1R ([^125^I]VIP IC_50_ = 100 nM), VPAC2R ([^125^I]VIP IC_50_ = 10 µM) ^1^	[144]
	[Lys^15^, Arg^16^, Leu^27^]-VIP(1-7)/GRF(8-27)	Agonist	Chimeric VIP/GRF analogue	VPAC1R ([^125^I]VIP IC_50_ = 1 nM), VPAC2R ([^125^I]VIP IC_50_ > 30 µM) ^2^	[144]
	PG 97-269	Antagonist	N-terminal modified VIP/GRF chimeric analogue	VPAC1R ([^125^I]VIP IC_50_ = 2 nM), VPAC2R ([^125^I]VIP IC_50_ = 3 μM)	[145]
VPAC2R	RO 25-1392	Agonist	Cyclic VIP analogue	VPAC1R ([^125^I]VIP K_i_ = 1 μM), VPAC2R ([^125^I]VIP K_i_ = 9.6 nM)	[146]
	RO 25-1553	Agonist	Cyclic VIP analogue	VPAC1R ([^125^I]VIP IC_50_ = 800 nM), VPAC2R ([^125^I]RO 25-1553 IC_50_ = 1 nM)	[147]
	PG 96-249	Agonist	Linear RO 25-1553 analogue	VPAC1R ([^125^I]VIP IC_50_ = 3 μM), VPAC2R ([^125^I]RO 25-1553 IC_50_ = 10 nM)	[147]
	BAY 55-9837	Agonist	PACAP/VIP analogue	PAC1R ([^125^I]PACAP27 IC_50_ = N/A) ^3^ VPAC1R ([^125^I]PACAP27 IC_50_ = 8.7 µM), VPAC2R ([^125^I]PACAP27 IC_50_ = 60 nM)	[34]
	PG 99–465	Antagonist	N-terminal myristoylated, C-terminal elongated VIP analogue	VPAC1R ([^125^I]VIP IC_50_ = 200 nM), VPAC2R ([^125^I]RO 25-1553 IC_50_ = 1 nM)	[147]
PAC1R	M65	Antagonist	Maxadilan analogue	PAC1R ([^125^I]PACAP27 K_d_ = 0.6 nM), VPAC1R ([^125^I]VIP K_d_ = N/A), VPAC2R ([^125^I]VIP K_d_ = N/A) ^4^	[133]
	max.D.4	Antagonist	Maxadilan analogue	PAC1R ([^125^I]PACAP27 K_d_ = 0.6 nM), VPAC1R ([^125^I]VIP K_d_ = N/A), VPAC2R ([^125^I]VIP K_d_ = N/A) ^4^	[148]
	PACAP(6-38)	Antagonist	N-terminal truncated PACAP analogue	PAC1R ([^125^I]Ac-His^1^-PACAP27 K_i_ = 30 nM), VPAC1R ([^125^I]VIP K_i_ = 600 nM), VPAC2R ([^125^I]Ac-His^1^-PACAP27 K_i_ = 40 nM) ^5^	[149,150]

* The radioligand used is specified for radioligand binding assays. ^1^ Human VPACRs and rat PAC1R used for the assay. ^2^ Selectivity for growth hormone-releasing factor (GRF) receptor was not tested. ^3^ For BAY 55-9837, no competitive binding was observed for PAC1R. ^4^ For max.D.4 and M65, competition of [^125^I]PACAP27 and [^125^I]VIP binding to the VPAC receptors was not observed. ^5^ Also displays significant affinity for VPAC2R [149,150,151].

**Table 3 ijms-23-08069-t003:** Deletion (grey) or modification (red) of the N-terminal residues in VIP and PACAP analogues can be used to generate receptor-selective peptide antagonists.

N-Terminal Truncation/Modification																																						
	1				5				10					15					20					25					30					35				
**PACAP38**	H	S	D	G	I	F	T	D	S	Y	S	R	Y	R	K	Q	M	A	V	K	K	Y	L	A	A	V	L	G	K	R	Y	K	Q	R	V	K	N	K
**PACAP(6–38)**	H	S	D	G	I	F	T	D	S	Y	S	R	Y	R	K	Q	M	A	V	K	K	Y	L	A	A	V	L	G	K	R	Y	K	Q	R	V	K	N	K
**VIP**	H	S	D	A	V	F	T	D	N	Y	T	R	L	R	K	Q	M	A	V	K	K	Y	L	N	S	I	L	N										
**PG 97-269 ^1^**	H	F	D	A	V	F	T	N	S	Y	R	K	V	L	K	R	L	S	A	R	K	L	L	Q	D	I	L											
**PG 99-465 ^2^**	H	S	D	A	V	F	T	D	N	Y	T	K	L	R	K	Q	M	A	V	K	K	Y	L	N	S	I	K	K	G	G	T							

^1^ The following non-canonical amino acid modifications observed in PG 97-269: Acetyl-His^1^ D-Phe^2^ [145]. ^2^ The following non-canonical amino acid modifications observed in PG 97-269: Myr-His^1^ [147].

**Table 4 ijms-23-08069-t004:** Truncation (grey) and elongation (red) peptide length contribute to VPAC1R/VPAC2R-selective peptide analogues.

C-Terminal Truncation/Elongation																																						
	1				5				10					15					20					25					30					35				
**PACAP38**	H	S	D	G	I	F	T	D	S	Y	S	R	Y	R	K	Q	M	A	V	K	K	Y	L	A	A	V	L	G	K	R	Y	K	Q	R	V	K	N	K
**[R^16^]-PACAP(1–23)**	H	S	D	G	I	F	T	D	S	Y	S	R	Y	R	R	Q	M	A	V	K	K	Y	L	A	A	V	L	G	K	R	Y	K	Q	R	V	K	N	K
**VIP**	H	S	D	A	V	F	T	D	N	Y	T	R	L	R	K	Q	M	A	V	K	K	Y	L	N	S	I	L	N										
**RO 25-1553 ^1^**	H	S	D	A	V	F	T	E	N	Y	T	K	L	R	K	Q	L	A	A	K	K	Y	L	N	D	L	K	K	G	G	T							
**PG 96-249a ^2^**	H	S	D	A	V	F	T	E	N	Y	T	K	L	R	K	Q	L	A	A	K	K	Y	L	N	D	L	K	K	G	G	T							

^1^ The following non-canonical amino acid modifications observed in RO 25-1553: Ac-His^1^ and Nle^17^. Lactam bridge formed between Lys^21^ and Asp^23^ [147]. ^2^ The following non-canonical amino acid modifications observed in PG 96-249a: Ac-His^1^ and Nle^17^ [147].

**Table 5 ijms-23-08069-t005:** List of structures of the ECD region of the PACAP subfamily of receptors determined through X-ray crystallography and NMR.

Receptor	Region	Agonist	PDB	Resolution	Method
VPAC2R [296] ^1^	ECD	–	2X57	2.1 Å	X-ray diffraction
PAC1sR [297]	ECD	–	3N94	1.8 Å	X-ray diffraction
PAC1sR [298]	ECD	PACAP (6–38)	2JOD	–	Solution NMR

^1^ Structure deposited and released on the RCSB protein databank (PDB) with no corresponding publication.

**Table 6 ijms-23-08069-t006:** Cryo-EM structures of active, peptide-bound, G_s_ protein-coupled, PAC1R, VPAC1R, and VPAC2R as of June 2022. The PAC1R isoforms solved include PAC1R-null (PAC1nR) with the full-length ECD and PAC1R-short (PAC1sR) with the truncated N-terminal ECD.

Receptor	Agonist	G Protein ^1^	PDB	Resolution
PAC1nR [135]	PACAP38	DNGα_s_, Gβ_1_, Gγ_2_	6P9Y	3.0 Å
PAC1nR [139]	PACAP38	Mini-Gα_s_, Gβ_1_, Gγ_2_	6LPB	3.9 Å
PAC1sR [138]	PACAP38	DNGα_s_, Gβ_1_, Gγ_2_	6M1I	3.5 Å
PAC1sR [138]	Maxadilan	DNGα_s_, Gβ_1_, Gγ_2_	6M1H	3.6 Å
VPAC1R-LgBiT ^2^ [136]	PACAP27	DNGα_s_, Gβ_1_-HiBiT, Gγ_2_	6VN7	3.2 Å
VPAC2R-LgBiT ^2^ [137]	PACAP27	DNGα_s_, Gβ_1_-HiBiT, Gγ_2_	7VQX ^3^	2.7 Å

^1^ Gα_s_ modifications to aid complex stability include the use of a dominant negative (DN)Gα_s_ and a mini-Gα_s_ [284,286]. ^2^ The VPAC receptors utilized NanoBiT tethering technology to aid complex stability where the receptor is tagged with a large BiT (LgBiT) component and the Gβ_1_ subunit is tagged with an engineered small BiT (HiBiT) component [136]. ^3^ In Xu et al., 2022 [137], two PACAP27-bound VPAC2R structures were determined. One with the N-terminal modifications, VPAC2R(24-438) (PDB: 7WBJ), and one with no N-terminal modifications, VPAC2R(1–438) (PDB: 7VQX). As the construct with the N-terminal modifications displayed an altered pharmacology profile, the construct with no N-terminal modifications (PDB: 7VQX) is the construct discussed in this review.

## Data Availability

Not applicable.

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
