# Peer review of "Targeting VIP and PACAP Receptor Signaling: New Insights into Designing Drugs for the PACAP Subfamily of Receptors"

_ijms, 2022, doi:10.3390/ijms23158069_

Round 1
Reviewer 1 Report
Dear Authors,
The topic is timely and may attract much attention. The authors have done a very detailed, thorough job.
I have only some suggestions to improve this paper:
1. It is not usual to use references in the abstract.
2. The authors could better detail the role of PACAP and VIP in diseases.
3. I may have missed it, but I haven't read much information about the current therapeutic options for PACAP and VIP.
Recommendation of revision: minor
Author Response
Report 1
- It is not usual to use references in the abstract.
Agree, we have removed the references from the abstract.
- The authors could better detail the role of PACAP and VIP in diseases.
Thanks for the suggestion, section 1 (Table 1) and section 2 (subsections 2.2.1 and 2.2.5) have been updated
- I may have missed it, but I haven't read much information about the current therapeutic options for PACAP and VIP.
The current therapeutic options are limited for PACAP and VIP and are mentioned in “Section 2.4. – Understanding of PACAP and VIP signaling and regulation that needs to be addressed for disease-focused therapies”: “These receptors remain attractive targets for inflammatory and immune diseases, and disorders in the nervous system. However, there are currently no drugs on the market that specifically target PACAP/VIP receptors. While PACAP has been implicated in migraine, current efforts to antagonise PACAP-PAC1R signaling through the development of monoclonal antibodies to PACAP38 or PAC1R for treatment-resistant migraines have not been successful. In phase II clinical trials, the PAC1R monoclonal antibody, AMG301 failed to exhibit efficacy for migraine treatment, despite promising results in preclinical studies.”
Reviewer 2 Report
The authors report the molecular mechanisms involved in the selectivity of VIP and PACAP receptor signaling. The review is divided into four parts: (1) a well-conceived introduction summarizing knowledge about VIP, PACAP, and their receptors (2) an extensive overview describing signaling pathways activated by PACAP subfamily receptors (3) a well-detailed section regarding the structural characteristics of VIP and PACAP receptors and (4) the conclusion section.
In general, the review is built up logically and provides interesting, up-to-date information on the latest achievements in the field. Besides the considerable merit, I have some concerns and suggestions regarding content and references.
1. In the first section "1. Introduction – Physiological roles of the PACAP and VIP receptors", the authors describe the role of the peptides and their expression in the central and peripheral nervous system as well as in different organs. However, several information regarding the expression and the role of these peptides in peripheral organs missing. For example, the authors should include data concerning the expression of PACAP receptors in the skeletal system, the hematopoietic organs, and the visual system (e.g. PMID: 26925806; PMID: 30548314; PMID: 23189073). Thus, table 1 should be completed.
2. In section “2.2.1 Alternative Splicing”, lines 247-249, the authors stated that PAC1R splice isoforms have distinct pharmacological profiles. It would be interesting if the authors also discussed the differential expression of the splice variants based on the considered tissue/organ.
3. In section "2.2.5. Additional Downstream Signal Transduction Pathways", the authors describe additional signaling pathways triggered by PACAP receptors. Here, they have to add a paragraph about the transactivation of epidermal growth factor receptor PACAP-mediated (e.g. PMID: 17332755; PMID: 31247223). Figure 1 has to be modified accordingly.
Author Response
Report 2
- In the first section "1. Introduction – Physiological roles of the PACAP and VIP receptors", the authors describe the role of the peptides and their expression in the central and peripheral nervous system as well as in different organs. However, several information regarding the expression and the role of these peptides in peripheral organs missing. For example, the authors should include data concerning the expression of PACAP receptors in the skeletal system, the hematopoietic organs, and the visual system (e.g. PMID: 26925806; PMID: 30548314; PMID: 23189073). Thus, table 1 should be completed.
Thank you for pointing this out, we agree with the comment. Table 1 in section 1 has been updated to include expression data on the visual system and the haematopoietic cells in the bone marrow
- In section “2.2.1 Alternative Splicing”, lines 247-249, the authors stated that PAC1R splice isoforms have distinct pharmacological profiles. It would be interesting if the authors also discussed the differential expression of the splice variants based on the considered tissue/organ.
Thank you for the suggestion, section 2.2.1 has been updated to include differential expression of the splice variants.
- In section "2.2.5. Additional Downstream Signal Transduction Pathways", the authors describe additional signaling pathways triggered by PACAP receptors. Here, they have to add a paragraph about the transactivation of epidermal growth factor receptor PACAP-mediated (e.g. PMID: 17332755; PMID: 31247223). Figure 1 has to be modified accordingly.
Section 2.2.5 has been updated to include a section of the transactivation of epidermal growth factor receptor and figure 1 has been updated accordingly.